# Prevalence and nature of self-reported visual complaints in people with Parkinson's disease—Outcome of the Screening Visual Complaints questionnaire

Iris van der Lijn[1,2]*, Gera A. de Haan[1,2], Fleur E. van der Feen[1,2], Famke Huizinga[1,3], Catharina Stellingwerf[2], Teus van Laar[4], Joost Heutink[1,2]

1 Department of Clinical and Developmental Neuropsychology, University of Groningen, Groningen, the Netherlands, 2 Royal Dutch Visio, Centre of Expertise for Blind and Partially Sighted People, Huizen, the Netherlands, 3 Department of General Practice and Elderly Care Medicine, University of Groningen, University Medical Centre Groningen, Groningen, The Netherlands, 4 Department of Neurology, University of Groningen, University Medical Centre Groningen, Groningen, the Netherlands

* i.van.der.lijn@rug.nl

## Abstract

### Introduction

Visual complaints can have a vast impact on the quality of life of people with Parkinson's disease (PD). In clinical practice however, visual complaints often remain undetected. A better understanding of visual complaints is necessary to optimize care for people with PD and visual complaints. This study aims at determining the prevalence of visual complaints experienced by a large outpatient cohort of people with PD compared to a control group. In addition, relations between visual complaints and demographic and disease-related variables are investigated.

### Methods

The Screening Visual Complaints questionnaire (SVCq) screened for 19 visual complaints in a cohort of people with idiopathic PD (n = 581) and an age-matched control group without PD (n = 583).

### Results

People with PD experienced significantly more complaints than controls, and a greater impact of visual complaints on their daily lives. Complaints that were most common ('often/always') were unclear vision (21.7%), difficulty reading (21.6%), trouble focusing (17.1%), and blinded by bright light (16.8%). Largest differences with controls were found for double vision, needing more time to see and having trouble with traffic participation due to visual complaints. Age, disease duration, disease severity, and the amount of antiparkinsonian medication related positively to the prevalence and severity of visual complaints.

**Data Availability Statement:** The data that support the findings of this study are available from

DataverseNL. Because of ethical reasons, restrictions apply to the availability of these data (i.e., the combination of variables can potentially lead to participants being identified). Data are available at https://doi.org/10.34894/ODVGUV with the permission of the ethics board of the Faculty of Behavioral and Social Sciences, University of Groningen (contact: dcc@rug.nl).

**Funding:** This work was supported by Visio Foundation, Amsterdam, The Netherlands (JH, GdH, FvdF, IvdL, CS; https://visiofoundation.org/), and ZonMw grant 637005001 (JH, GdH, FvdF, IvdL, CS; Expertisefunctie Zintuiglijk Gehandicapten, Meerjarig deelsectorplan 2020-2022 Visueel; https://www.zonmw.nl/nl/). The funders had no role in study design, data collection and analysis, decision to publish, or preparation of the manuscript.

**Competing interests:** The authors have declared that no competing interests exist.

## Conclusion

Visual complaints are highly prevalent and occur in great variety in people with PD. These complaints progress with the disease and have a large impact on the daily lives of these people. Standardized questioning is advised for timely recognition and treatment of these complaints.

## Introduction

Visual problems are increasingly recognized as part of the symptomatology of Parkinson's disease (PD) [1]. The presence of visual problems may have a vast impact on the quality of life of people with PD [2,3]. Vision is essential for the performance of most daily life activities, e.g., reading or mobility-related tasks such as driving. Moreover, people with PD may need to rely on vision even more than healthy individuals do, to compensate for the loss in motor function [4]. Loss of visual function may thus increase fall risk and decrease independence in daily life [3–5].

Timely recognition and treatment of visual complaints is therefore of high importance. As self-reported complaints are an indication of the difficulties a person encounters in daily life, asking about them is an essential complement to screening for possible ophthalmological conditions (OCs) or visual function disorders. It ensures that treatment is focused on the person's priorities, regardless of the cause [6,7].

However, in scientific literature and clinical practice, visual complaints of people with PD are often underreported and unrecognized [4,8]. A recent systematic review on visual complaints in people with PD indicated that literature on this topic is still scarce [2]. Most studies had small sample sizes and examined only one or few visual complaints. Literature did indicate that visual complaints occur in great variety in people with PD. Complaints were described in terms of functions (e.g., blurred vision, double vision, increased sensitivity to light or changes in contrast sensitivity) or activities (e.g., difficulty reading, reaching, or driving). Most studies included in this review showed that visual complaints were more prevalent in people with PD compared to people without PD, and that the presence of visual complaints was related to a longer disease duration, higher disease severity, and off-state.

In clinical practice, visual complaints often go undetected for a number of reasons. People with PD might not report their visual problems because, e.g., a) other symptoms may be more prominent, b) there is limited time during health care visits to discuss all problems, c) people tend to find it hard to specify their visual complaints [9], d) people might not connect visual problems to PD, since this connection is often unknown [3] and visual problems might present even before motor symptoms arise [10], and e) in most cases no structured questions are asked by the clinician, resulting in a full reliance on the person's (possibly decreased) self-initiated declaration [11]. A better understanding of PD-related visual complaints is important to guide care, thereby optimizing the safety and quality of life of people with PD [3]. This can be achieved by studying a diversity of function and activity related visual complaints in a large sample, which has not been done before. The Screening Visual Complaints questionnaire (SVCq) was developed for this purpose and validated in a community sample (individuals without severe ophthalmological, neurological and psychiatric disorders) [12]. To validate the SVCq in a clinical sample, our previous study examined the factor structure of the questionnaire, based on data from a large cohort of people with PD and an age-matched control group randomly selected from the aforementioned community sample [13]. A five-factor model (i.e.,

five subscales) was identified for good understanding of the distribution of complaints in people with PD. However, to provide a straightforward indication of the presence of visual complaints or to identify specific targets for treatment, the previous study indicated that in addition to subscale scores, total scores or scores on individual items might also be useful [13]. The current study therefore uses these scores to investigate the prevalence of self-reported visual complaints in people with PD, compared to a control group, using the validated SVCq on the same samples as the previous study [13]. In addition, the relationship between these complaints and various demographic and disease-related characteristics and the presence of an identified OC will be explored.

## Method

This study used a large dataset of people with PD and controls, which is the same dataset as used in the study regarding the factor structure of the SVCq [13].

### Participants

All Dutch speaking people with PD who visited the Parkinson Expertise Center in Groningen, the Netherlands, were eligible for participation in the study. PD diagnosis was based on the UK Parkinson's Disease Society Brain Bank Diagnostic Criteria [14]. Data of a comparison group without PD was collected by Huizinga et al. (2020) [12]. This group was selected to be representative of the community sample, without severe self-reported neurological, ophthalmological, and psychiatric conditions; mild conditions were allowed in this group (see notes e–f below Table 1). See our previous study for more details on inclusion and participants [13].

### Materials

**Screening visual complaints questionnaire.** The SVCq screens for visual complaints in people with neurodegenerative diseases [12]. The first item of the SVCq inquires individuals whether they experienced any visual complaints in the past couple of weeks, and if so, what kind of visual complaints they have experienced. This open-ended question allows people to report spontaneous complaints. This question is followed by 19 structured items, each describing a visual complaint. Individuals indicated the frequency in which they experienced each complaint ('never/hardly', 'sometimes', 'often/always'). In addition, to measure the impact of the visual complaints on their daily lives, individuals indicated to what extent the complaints limited them in their daily lives (on a scale from 0 (not limited at all) to 10 (severely limited); i.e., 'impact on daily life score'). In addition to the visual complaints, individuals were asked to provide demographics (i.e., year of birth, age, gender, and level of education) and any present OC. In individuals without severe ophthalmological, neurological and psychiatric disorders, the SVCq has shown a good internal consistency ($\alpha = 0.85$) and test-retest reliability (ICC = 0.82) [12]. In addition, a clear and clinically relevant factor structure has been demonstrated in people with PD; five subscales were identified. Together with the total score, the impact on daily live score, and scores on individual items of the questionnaire, the five subscales contribute to a full understanding of a person's complaints, providing a solid basis for individualized visual care [13].

**Data from medical files.** Disease-related data were recorded from the medical files of each individual, including disease type (PD or another related disorder), duration (time since diagnosis) and Hoehn and Yahr stage (H&Y) [15] for disease severity. Furthermore, data on current medication, neurological, ophthalmological, or psychiatric comorbidity, as well as a history of Deep Brain Stimulation (DBS) were collected.

**Table 1. Demographics and disease characteristics of people with PD and age-matched controls.**

| | | People with PD | Controls |
|---|---|---|---|
| N | | 581 | 583 |
| Sex (n, % female) | | 227, 39.1% | 214, 36.7% |
| Age (years; M ± SD) | | 69.25 ± 9.01 | 69.17 ± 8.99 |
| Education[a] (n, %) | | | |
| | Low | 100, 17.2% | 132, 22.7% |
| | Medium | 211, 36.3% | 146, 25.1% |
| | High | 265, 45.6% | 303, 52.2% |
| Disease duration (years; M ± SD) | | 7.96 ± 6.59 | - |
| H&Y stage (n, %) | | | |
| | 1 | 125, 21.5% | - |
| | 2 | 218, 37.5% | - |
| | 3 | 101, 17.4% | - |
| | ≥4 | 49, 8.4% | - |
| | Missing | 88, 15.2% | - |
| Presence of DBS (n, %) | | 81, 13.9% | - |
| LEDD[b] (mg; M ± SD); missing (n, %) | | 907.75 ± 592.01; 5, 0.9% | - |
| | Levodopa (n, %) | 561, 97.4% | |
| | Dopamine agonists (n, %) | 264, 45.8% | |
| | Monoamine oxidase-B inhibitors (n, %) | 28, 4.9% | |
| | No antiparkinsonian medication (n, %) | 6, 1.0% | |
| Visited ophthalmologist (self-reported) | | 253, 43.5% | 243, 41.7% |
| Presence of severe neurological condition (n, %)[c] | | 51, 8.8%[d] | - |
| Presence of severe psychiatric condition (n, %)[c] | | 13, 2.2%[e] | - |
| Presence of any ophthalmological condition (n, %)[f] | | | |
| | Yes | 203, 34.9% | 127, 21.8% |
| | No | 351, 60.4% | 407, 69.8% |
| | Unclear | 27, 4.7% | 49, 8.4% |

*Note*: This table is similar to the one describing the same population in a previous publication [13]. DBS = Deep Brain Stimulation; H&Y = Hoehn and Yahr staging [15]; LEDD = Levodopa equivalent daily dose; M = mean; mg = milligram; n = number; PD = Parkinson's disease; SD = standard deviation.

[a] Categorization based on the International Standard Classification of Education (ISCED) [16].

[b] LEDD calculated according to protocol of Tomlinson et al. (2010) [17].

[c] Severe conditions that were used as an exclusion criterion for controls and might influence vision.

[d] Cerebrovascular accident (n = 16), transient ischemic attack (n = 15), epilepsy (n = 10), basilar skull fracture/traumatic injury (n = 6), thalamatomy (n = 4), encephalopathy (n = 2), brain tumor (n = 2), neuroborreliosis (n = 1), cavernoma (n = 1), and pituitary tumor resection (n = 1).

[e] Schizophrenia/psychosis (n = 13).

[f] See S1 Table.

## Procedure

All Dutch speaking patients visiting the Parkinson Expertise Center in Groningen between May 1st 2019 and February 3rd 2021 were asked to fill out the SVCq. The SVCq is freely available for use in clinical practice (see supplementary material or the original publication [12]: for the Dutch version, see S2 Table or https://doi.org/10.1371/journal.pone.0232232.s001 [12]; for the English version, see S3 Table or https://doi.org/10.1371/journal.pone.0232232.s002 [12]). They were asked to complete the questionnaire in the waiting room. Individuals who were not able to complete the questionnaire on location were provided with a pre-paid return envelope to complete the questionnaire at home. Another option was to complete the SVCq online via

Qualtrics (https://www.qualtrics.com). Due to COVID restrictions, from March 2020 on, SVCq distribution was done by sending people questionnaires to fill out at home.

In addition to the questionnaire, people could indicate whether they were interested to receive advice or rehabilitation for their visual complaints.

All individuals signed informed consent for providing pseudo-anonymized questionnaire data and several disease-related characteristics from their medical charts for research. According to the Medical Ethics Committee of the University Medical Center Groningen, no further ethical approval from the committee was required as all data were collected from standard care.

Data of controls were collected through Panel Inzicht, an online research panel in the Netherlands. A small financial reward was provided to individuals for filling out the online version of the SVCq. All participants provided written informed consent and ethical approval was provided by the Ethical Committee Psychology of the University of Groningen. See previous studies for more details on data collection and procedures [12,13].

## Data-analysis

All analyses were performed with SPSS 26 [18]. Data used in this study are available upon request via DataverseNL (https://doi.org/10.34894/ODVGUV).

**Matching.** The control group was matched to the group of people with PD in terms of age. The PD group was split into several age groups with a 5-year range. The number of controls per age group was based on the percentage of people with PD in each age group. The highest possible number of controls in each age group was randomly selected from the total group of controls.

**Handling missing or incorrect data.** There were no missing data in the control group. PD data were removed case wise if missing answers exceeded 25% of SVCq items (n = 5). Missing answers that remained (0.2% of total data) were left missing for the frequency analysis of visual complaints. To be able to calculate total SVCq scores, missing values were imputed based on all available values using the Maximum Likelihood Estimation method. In ten PD cases, two answer boxes were checked on one of the 19 items. The answer representing the lowest frequency of visual complaints was preserved, in order to prevent an overestimation of visual complaints.

**Determining the presence of ophthalmological conditions.** We considered people with PD to be free of an OC if their medical file and their self-reported information were not suggestive of an OC. If they did report a clear condition or there was an OC mentioned in the medical file, they were considered having an OC. In the control group, the presence of an OC was based only on self-reported information. See S1 Table for an overview of OCs in both groups. We considered people who underwent cataract surgery (N = 38 in people with PD and N = 43 in the control group) to have an OC, given the possibility of residual long-term effects or secondary cataracts. Excluding these people did not alter the results.

**Assumptions.** All variables were either on interval (total score, number of complaints, impact on daily life (0–10 scale)) or ordinal level (scores on individual complaints) and measures were independent. The Kolmogorov-Smirnov tests showed that normality was violated for all variables. Therefore, non-parametric tests were used.

**Prevalence of visual complaints.** A frequency analysis was performed for people with PD and controls on each of the 19 complaints of the SVCq; percentages were calculated of persons answering 'never/hardly', 'sometimes', and 'often/always'. Frequencies were compared between people with PD and controls using a Chi-Square test, complemented with adjusted residuals. Comparisons were made for each complaint. Taking multiple comparisons into

account, Bonferroni's correction resulted in an alpha of .008 (.05/6 = .008) and adjusted residuals of $\geq$ 2.64 to reach significance [19]. Effect sizes of Cramer's V were calculated (small: .07 - .21, medium: .21 - .35, large: > .35) [20].

An additional frequency analysis was performed on the answers of people with PD to the first, open-ended question of the SVCq. These spontaneous complaints were categorized according to complaints listed in the structured part of the SVCq. If a complaint was different from SVCq complaints, a new category was created when the complaint was mentioned at least twice. When only mentioned once, complaints were added to the category 'Other'. Complaints were excluded if they were too unclear for interpretation (n = 9), evidently unrelated to vision (n = 3), referred to an OC (e.g., glaucoma; n = 17), or to having or needing glasses/contact lenses (n = 34). Two researchers were involved in the categorization process. In case of doubt, a third researcher was involved.

**The number and impact of visual complaints and the relationship to other variables.** SVCq total score, the impact on daily life score (0–10 scale), the total number of complaints ('sometimes' and 'often/always'), and the number of complaints rated 'often/always' were calculated for all people with PD and controls. The total SVCq score was calculated by summing the numbers of each response to the 19 items (0 = 'never/hardly', 1 = 'sometimes', 2 = 'often/always'). The relationship between these scores and age, disease duration and total Levodopa Equivalent Daily Dose (LEDD) of the PD group was calculated by Spearman's correlations. In addition, Kruskal-Wallis and Mann-Whitney-U tests were performed to investigate differences in these scores between 1) people with PD and controls, 2) people with PD with and without an OC, and 3) people with PD and controls without an OC, 4) male and female people with PD, and 5) people with PD in different disease severity groups (H&Y 1, H&Y 2, H&Y 3, and $\geq$ H&Y 4). An alpha < .05 was considered to be significant. Coefficient r was calculated as an effect size (small: .1 - .3, medium: .3 - .5, large: .5–1.0) [21]. See our previous publication for results regarding the five subscales of the SVCq [13].

## Results

In total, 614 patients filled out the SVCq, of which 33 did not meet the diagnosis of idiopathic PD. These people were excluded, leaving a PD group of 581 individuals. Selected from a group of 1402 controls, the matching procedure resulted in a control group of 583 individuals. Characteristics of both groups are presented in Table 1 (see also [13]).

### Prevalence of visual complaints

Table 2 presents the prevalence of visual complaints in people with PD and controls. More than 90% of people with PD reported at least one complaint ('sometimes' or 'often/always'; 90.7%, n = 527), and 61.3% (n = 356) reported five or more complaints. Of the controls, the majority also reported at least one complaint (85.9%, n = 501), and 39.3% reported five or more. The prevalence of the 19 visual complaints ('often/always') ranged from 2.1% to 21.7% in people with PD and from 0.3% to 11.0% in controls. Most frequently reported complaints in people with PD were unclear vision and trouble reading (> 20%), followed by trouble focusing, experiencing reduced contrast, being blinded by bright light, and needing more light (> 15%). Least common were complaints of shaky, jerky, shifting images, visual field, color vision, and painful eyes (< 5%). Nearly half of people with PD (42.3%, n = 246) indicated that they would appreciate advice or rehabilitation for their visual complaints.

All complaints were significantly more prevalent in people with PD compared to controls, except for complaints on light/dark adjustment, color vision and painful eyes. Based on adjusted residuals, differences between the two groups were found primarily in the response

**Table 2. Overview of the frequencies of people with PD and controls experiencing the SVCq's visual complaints, and results of Chi-square tests comparing the groups.**

| Complaint | | People with PD | Controls | $X^2$ | p | Cramer's V |
|---|---|---|---|---|---|---|
| | | Prevalence (adjusted residuals) | | | | |
| Unclear vision | | | | 29.14 | < .001* | .16 |
| | Often/always | 21.7% (5.4*) | 10.1% (-5.4*) | | | |
| | Sometimes | 39.8% (-2.1) | 46.0% (2.1) | | | |
| | Never/hardly | 38.6% (-1.9) | 43.9% (1.9) | | | |
| Trouble focusing | | | | 43.77 | < .001* | .19 |
| | Often/always | 17.1% (6.4*) | 5.3% (-6.4*) | | | |
| | Sometimes | 34.7% (0.1) | 34.3% (-0.1) | | | |
| | Never/hardly | 48.3% (-4.1*) | 60.4% (4.1*) | | | |
| Double vision | | | | 73.83 | < .001* | .25 |
| | Often/always | 10.8% (6.1*) | 2.1% (-6.1*) | | | |
| | Sometimes | 19.3% (5.4*) | 8.4% (-5.4*) | | | |
| | Never/hardly | 69.9% (-8.3*) | 89.5% (8.3*) | | | |
| Depth perception | | | | 29.61 | < .001* | .16 |
| | Often/always | 10.2% (3.9*) | 4.3% (-3.9*) | | | |
| | Sometimes | 25.4% (3.2*) | 17.7% (-3.2*) | | | |
| | Never/hardly | 64.4% (-5.1*) | 78.0% (5.1*) | | | |
| Shaky, jerky, shifting images | | | | 18.73 | < .001* | .13 |
| | Often/always | 2.1% (2.7*) | 0.3% (-2.7*) | | | |
| | Sometimes | 12.6% (3.3*) | 6.9% (-3.3*) | | | |
| | Never/hardly | 85.4% (-4.1*) | 92.8% (4.1*) | | | |
| Visual field | | | | 15.19 | .001* | .11 |
| | Often/always | 3.4% (3.6*) | 0.5% (-3.6*) | | | |
| | Sometimes | 10.0% (1.4) | 7.7% (-1.4) | | | |
| | Never/hardly | 86.6% (-2.9*) | 91.8% (2.9*) | | | |
| Color vision | | | | 1.29 | .525 | .03 |
| | Often/always | 2.2% (0.9) | 1.5% (-0.9) | | | |
| | Sometimes | 5.2% (0.7) | 4.3% (-0.7) | | | |
| | Never/hardly | 92.6% (-1.1) | 94.2% (1.1) | | | |
| Reduced contrast | | | | 27.76 | < .001* | .16 |
| | Often/always | 16.1% (5.3*) | 6.4% (-5.3*) | | | |
| | Sometimes | 33.9% (-1.8) | 38.9% (1.8) | | | |
| | Never/hardly | 50.1% (-1.6) | 54.7% (1.6) | | | |
| Blinded by bright light | | | | 10.75 | .005* | .10 |
| | Often/always | 16.8% (2.8*) | 11.0% (-2.8*) | | | |
| | Sometimes | 31.3% (-2.4) | 37.9% (2.4) | | | |
| | Never/hardly | 52.0% (0.3) | 51.1% (-0.3) | | | |
| Needing more light | | | | 57.24 | < .001* | .22 |
| | Often/always | 16.6% (5.3*) | 6.7% (-5.3*) | | | |
| | Sometimes | 29.3% (4.2*) | 18.7% (-4.2*) | | | |
| | Never/hardly | 54.1% (-7.3*) | 74.6% (7.3*) | | | |
| Light/dark adjustment | | | | 7.48 | .024 | .08 |
| | Often/always | 8.8% (2.4) | 5.2% (-2.4) | | | |
| | Sometimes | 25.7% (0.9) | 23.5% (-0.9) | | | |
| | Never/hardly | 65.6% (-2.1) | 71.4% (2.1) | | | |
| Seeing things that others do not | | | | 53.44 | < .001* | .21 |

*(Continued)*

**Table 2.** (Continued)

| Complaint | | People with PD | Controls | $X^2$ | $p$ | Cramer's V |
|---|---|---|---|---|---|---|
| | | Prevalence (adjusted residuals) | | | | |
| | Often/always | 7.9% (5.0*) | 1.7% (-5.0*) | | | |
| | Sometimes | 24.8% (4.8*) | 13.7% (-4.8*) | | | |
| | Never/hardly | 67.2% (-6.9*) | 84.6% (6.9*) | | | |
| Distorted images | | | | 36.70 | < .001* | .18 |
| | Often/always | 4.3% (4.0*) | 0.7% (-4.0*) | | | |
| | Sometimes | 10.4% (4.4*) | 3.8% (-4.4*) | | | |
| | Never/hardly | 85.3% (-5.9*) | 95.5% (5.9*) | | | |
| Painful eyes | | | | 7.81 | .020 | .08 |
| | Often/always | 2.3% (1.6) | 1.0% (-1.6) | | | |
| | Sometimes | 15.9% (2.2) | 11.5% (-2.2) | | | |
| | Never/hardly | 81.9% (-2.7*) | 87.5% (2.7*) | | | |
| Dry eyes | | | | 17.17 | < .001* | .12 |
| | Often/always | 11.9% (4.1*) | 5.2% (-4.1*) | | | |
| | Sometimes | 25.2% (-0.5) | 26.4% (0.5) | | | |
| | Never/hardly | 62.9% (-2.0) | 68.4% (2.0) | | | |
| Needing more time | | | | 68.56 | < .001* | .24 |
| | Often/always | 11.7% (7.0*) | 1.5% (-7.0*) | | | |
| | Sometimes | 30.3% (3.3*) | 21.8% (-3.3*) | | | |
| | Never/hardly | 58.0% (-6.8*) | 76.7% (6.8*) | | | |
| Traffic | | | | 63.09 | < .001* | .23 |
| | Often/always | 6.1% (6.7*) | 1.0% (-6.7*) | | | |
| | Sometimes | 13.0% (3.5*) | 6.7% (-3.5*) | | | |
| | Never/hardly | 81.0% (-6.9*) | 92.3% (6.9*) | | | |
| Looking for something | | | | 36.73 | < .001* | .18 |
| | Often/always | 11.1% (4.6*) | 1.5% (-4.6*) | | | |
| | Sometimes | 20.5% (3.6*) | 12.9% (-3.6*) | | | |
| | Never/hardly | 68.5% (-5.7*) | 85.6% (5.7*) | | | |
| Reading | | | | 35.85 | < .001* | .18 |
| | Often/always | 21.6% (5.2*) | 10.3% (-5.2*) | | | |
| | Sometimes | 29.3% (1.4) | 25.7% (-1.4*) | | | |
| | Never/hardly | 49.1% (-5.1*) | 64.0% (5.1*) | | | |

*Note*: PD = Parkinson's disease.

\* = significant adjusted residual (> 2.64) or a significant p-value ($\alpha$ < .008).

options 'often/always' and 'never/hardly'. People with PD were more likely to rate their complaints 'often/always', while controls were more likely to rate their complaints 'never/hardly'. Medium effect sizes were found for the complaints needing more light, needing more time, trouble with traffic participation, double vision, and seeing things that others do not. Other effect sizes were small. Fig 1 visually presents the prevalence of each visual complaint ('often/always') in both groups.

Analysis of responses to the open-ended question of the SVCq revealed that 52% (n = 299) of people with PD reported at least one spontaneous visual complaint (544 complaints in total). Most complaints were similar to structured SVCq items (79%; see Table 3). These complaints were also the most prevalent; i.e., unclear vision (19.3%, n = 112) and difficulty reading (15.0%, n = 87) were most frequently reported, followed by double vision (11.2%, n = 65) and

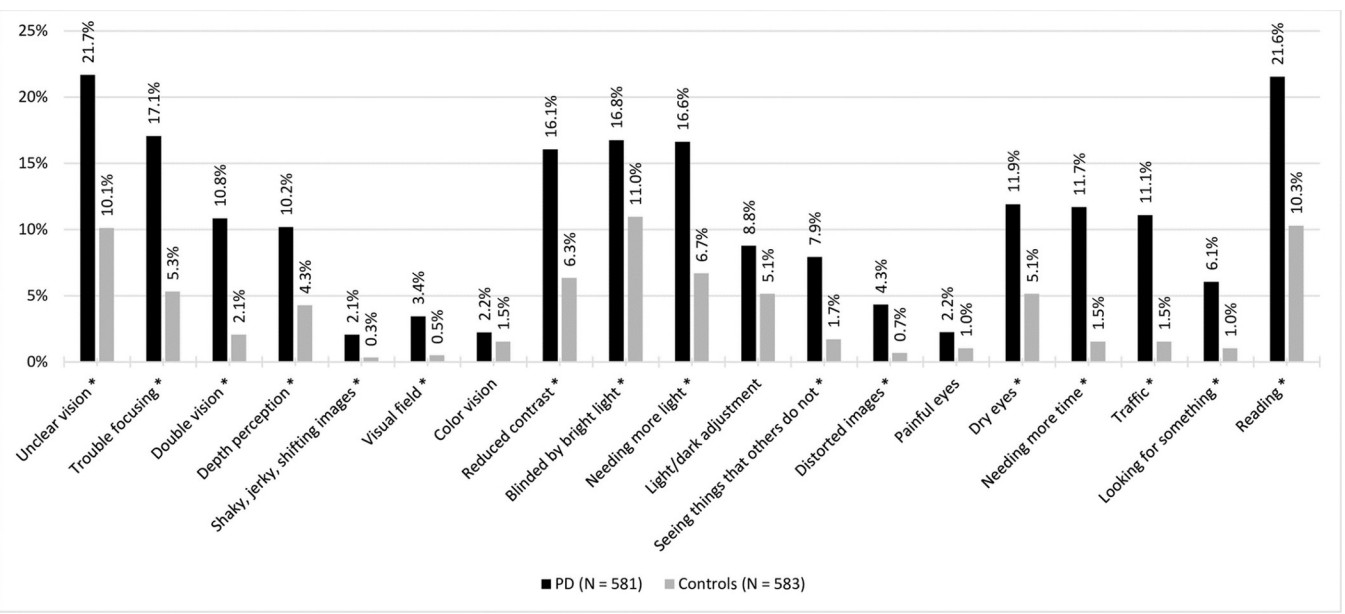

**Fig 1. Prevalence of visual complaints reported 'often/always' by people with Parkinson's disease and age-matched controls.** * = significant difference between the groups ($\alpha < .008$).

needing more light (6.0%, n = 35). Complaints dissimilar to SVCq items were also reported, though much less frequently; i.e., most frequently reported complaints were difficulty watching a display/TV (3.8%, n = 22) and tiredness while doing visual tasks (3.8%, n = 22).

## The number and impact of visual complaints and the relationship to other variables

People with PD reported significantly more complaints on the SVCq than controls (see Table 4). They had a higher total score, and a higher number of complaints ('sometimes' & 'often/always', and 'often/always'). In addition, they reported a greater impact of visual complaints on their daily lives. Effect sizes were all small.

Table 5 shows that people with PD with an OC reported significantly more complaints than those without an OC on all SVCq measures. In addition, a significant difference was found for most SVCq measures between people with PD without an OC and controls without an OC. No significant differences were found between these groups with regard to the impact on daily life experienced due to the presence of visual complaints (0–10 scale). Effect sizes for all comparisons were small.

The Kruskal-Wallis test performed on people with PD in different H&Y stages showed that all SVCq measures differed significantly between the groups (see Table 6A). The Mann-Whitney U comparing individual groups (see Table 6B) resulted in multiple differences, all indicating that people with PD in higher H&Y stages experienced more prevalent and more severe complaints. There were no differences between H&Y stage 1 and 2. Comparisons between other groups (1 vs. 3, 2 vs. 3, and 3 vs. ≥ 4) all revealed some significant differences. The comparisons 1 vs. ≥ 4 and 2 vs. ≥ 4 revealed significant differences for all SVCq measures. The total SVCq score, and the number of complaints ('sometimes' & 'often/always') showed significant differences in all comparisons (except 1 vs. 2). Most effect sizes were small. Medium effect sizes were found for comparisons of the groups 1, 2, and 3 with group ≥ 4 regarding the SVCq

**Table 3. Visual complaints reported spontaneously on the first, open-ended question of the SVCq by people with PD.**

| Spontaneous complaints similar to structured SVCq items, % (n) | | Spontaneous complaints not similar to structured SVCq items, % (n) | |
|---|---|---|---|
| Unclear vision | 23.2% (135) | Difficulty watching a display/TV | 3.8% (22) |
| Reading | 15.0% (87) | Tiredness during visual tasks | 3.8% (22) |
| Double vision | 11.2% (65) | Difficulty with distant vision | 3.3% (19) |
| Needing more light | 6.0% (35) | Vision varies during the day | 2.6% (15) |
| Trouble focusing | 3.4% (20) | Tearing of the eyes | 1.7% (10) |
| Traffic | 3.1% (18) | Eyelids close unwillingly | <1% (5) |
| Seeing things that others do not | 2.9% (17) | Difficulty seeing details/small things | <1% (3) |
| Painful eyes | 1.9% (11) | Difficulty with near vision | <1% (2) |
| Depth perception | 1.7% (10) | Itchy eyes | <1% (2) |
| Visual field | 1.4% (8) | Squeezing/squinting the eyes | <1% (2) |
| Blinded by bright light | 1.2% (7) | Other | 2.1% (12) |
| Reduced contrast | 1.0% (6) | | |
| Shaky, jerky, shifting images | <1% (4) | | |
| Distorted images | <1% (2) | | |
| Dry eyes | <1% (2) | | |
| Looking for something | <1% (2) | | |
| Needing more time | <1% (1) | | |
| **Total** | 79% (430) | | 21% (114) |

*Note*: PD = Parkinson's disease; SVCq = Screening Visual Complaints questionnaire.

total score and the number of complaints ('often/always'). Most medium effect sizes were found for the comparison between group 1 and $\geq 4$.

Most measures did not indicate a difference between the sexes (see Table 7). Females did experience complaints significantly more frequently 'often/always'. However, effect sizes were small. Age, disease duration, and LEDD showed significant positive correlations with all SVCq measures (see Table 8). Correlations were all weak [22].

**Table 4. SVCq scores of people with PD and the control group, with Mann-Whitney U test results.**

| | People with PD (n = 581) | | Controls (n = 583) | | | | |
|---|---|---|---|---|---|---|---|
| | M ± SD | Median | M ± SD | Median | U | p | r |
| Total SVCq score | 8.42 ± 7.27 | 7.00 | 5.19 ± 5.13 | 4.00 | 124430.5 | < .001* | 0.23 |
| Impact on daily life score | 3.16 ± 2.68 | 3.00 | 2.65 ± 2.52 | 2.00 | 148938.5 | .002* | 0.09 |
| N complaints 'sometimes' & 'often/always' | 6.37 ± 4.67 | 6.00 | 4.43 ± 3.80 | 4.00 | 128025.5 | < .001* | 0.21 |
| N complaints 'often/always' | 2.02 ± 3.04 | 1.00 | 0.76 ± 1.79 | 0.00 | 126403.5 | < .001* | 0.25 |

*Note*: M = mean; n = number; PD = Parkinson's disease; SD = standard deviation.

* = significant p-value ($\alpha < .05$).

**Table 5. SVCq scores of people with and without an ophthalmological condition, with Mann-Whitney U test results.**

| | PD OC+ (n = 203) | | PD OC- (n = 351) | | Control OC- (n = 407) | | PD OC+ vs. PD OC- | | | PD OC- vs. Control OC- | | |
|---|---|---|---|---|---|---|---|---|---|---|---|---|
| | M ± SD | Median | M ± SD | Median | M ± SD | Median | U | p | r | U | p | r |
| Total SVCq score | 10.33 ± 8.16 | 8.00 | 7.32 ± 6.59 | 6.00 | 4.61 ± 4.32 | 4.00 | 27760.0 | < .001* | 0.20 | 55159.0 | < .001* | 0.24 |
| Impact on daily life score | 3.77 ± 2.94 | 3.00 | 2.78 ± 2.47 | 2.00 | 2.52 ± 2.48 | 2.00 | 27832.0 | < .001* | 0.18 | 66161.0 | .148 | 0.05 |
| N complaints 'sometimes' & 'often/always' | 8.50 ± 4.10 | 8.00 | 5.74 ± 4.48 | 5.00 | 4.04 ± 3.52 | 3.00 | 22637.5 | < .001* | 0.29 | 56189.0 | < .001* | 0.20 |
| N complaints 'often/always' | 2.73 ± 3.11 | 2.00 | 1.58 ± 2.58 | 0.00 | 0.57 ± 1.14 | 0.00 | 25636.5 | < .001* | 0.20 | 56132.0 | < .001* | 0.25 |

*Note*: M = mean; n = number; OC+ = people with an ophthalmological condition; OC- = people without an ophthalmological condition; PD = Parkinson's disease; SD = standard deviation.

* = significant p-value (α < .05).

## Discussion

In this study, we investigated the prevalence of a wide variety of function and activity related visual complaints in a large group of people with PD, compared with an age-matched control group. Additionally, the relationship with several demographic and disease-related factors was examined.

Visual complaints were more prevalent in people with PD compared to controls, consistent with previous findings [2]. All 19 complaints were more prevalent in people with PD compared to controls, and 16 complaints showed significant differences. In most cases, the frequency of controls experiencing complaints 'sometimes' was comparable to people with PD. However, people with PD experienced complaints more frequently 'often/always'. This finding is important considering that the frequency in which a complaint is reported is closely related to its impact in daily life [23]. Furthermore, in contrast to controls, people with PD often reported more complaints concurrently. This might explain that people with PD experienced a greater impact of visual complaints on their daily lives (scale 0–10) compared to controls.

Most common in people with PD were complaints regarding unclear vision, reading, and trouble focusing. This is in line with previous literature indicating that blurred vision and reading difficulties were one of the most common visual complaints in people with PD [2]. Complaints of altered color vision and visual field deficits were less prevalent in previous studies [2]. Our study confirms these results, where these complaints, along with painful eyes and shaky, jerky, shifting, and distorted images, were least common. The largest differences between people with PD and controls were related to the complaints needing more light, needing more time, traffic participation, double vision, and seeing things that others do not. It can be concluded that these complaints are more unique to people with PD. The fact that these complaints stand out might be the reason that there is already more attention for some of these complaints than there is for others. For example, there are numerous studies on hallucinations [24], and double vision and oversensitivity to light are incorporated in widely used questionnaires [25–27]. However, even though other complaints stand out less, these are still highly prevalent and might be as bothersome in daily life [2].

The presence of an identified (diagnosed or self-reported) OCs related to the prevalence of visual complaints, evident from the differences found between people with PD with and without identified OCs. However, we showed that even without identified OCs, people with PD experienced more visual complaints than controls. We do have to mention that the presence of OCs may have been underestimated. Medical records and self-reported information of people with PD may not fully reflect the OCs present, for example because OCs may be

**Table 6. a. SVCq scores of people with PD in different disease severity stages, with Kruskal-Wallis test results. b. SVCq scores of people with PD in different disease severity stages, with Mann-Whitney U test results.**

| | H&Y 1 (n = 125) | | H&Y 2 (n = 218) | | H&Y 3 (n = 101) | | H&Y ≥ 4 (n = 49) | | H | df | p |
|---|---|---|---|---|---|---|---|---|---|---|---|
| | M ± SD | Median | M ± SD | Median | M ± SD | Median | M ± SD | Median | | | |
| Total SVCq score | 7.66 ± 7.38 | 6.00 | 7.56 ± 6.61 | 6.00 | 8.94 ± 6.59 | 7.00 | 13.54 ± 7.83 | 14.00 | 29.40 | 3 | < .001* |
| Impact on daily life score | 2.78 ± 2.57 | 2.00 | 3.05 ± 2.64 | 2.00 | 3.33 ± 2.56 | 3.00 | 4.69 ± 2.53 | 5.00 | 19.29 | 3 | < .001* |
| N complaints 'sometimes' & 'often/always' | 5.86 ± 4.79 | 5.00 | 5.86 ± 4.23 | 5.00 | 6.94 ± 4.48 | 6.00 | 9.39 ± 4.80 | 9.00 | 24.97 | 3 | < .001* |
| N complaints 'often/always' | 1.78 ± 3.01 | 0.00 | 1.69 ± 2.86 | 0.00 | 1.96 ± 2.51 | 1.00 | 4.08 ± 3.56 | 4.00 | 30.78 | 3 | < .001* |

| | H&Y 1 vs. H&Y 2 | | | H&Y 1 vs. H&Y 3 | | | H&Y 1 vs. H&Y ≥ 4 | | | H&Y 2 vs. H&Y3 | | | H&Y 2 vs. H&Y ≥ 4 | | | H&Y 3 vs. H&Y ≥ 4 | | |
|---|---|---|---|---|---|---|---|---|---|---|---|---|---|---|---|---|---|---|
| | U | p | r | U | p | r | U | p | r | U | p | r | U | p | r | U | p | r |
| Total SVCq score | 13223.5 | .649 | 0.03 | 5318.0 | .041* | 0.14 | 1682.0 | < .001* | 0.35 | 9440.5 | .040* | 0.11 | 2902.5 | < .001* | 0.31 | 1628.5 | .001* | 0.28 |
| Impact on daily life score | 13201.0 | .591 | 0.05 | 5294.0 | .087 | 0.12 | 1732.5 | < .001* | 0.32 | 9929.5 | .300 | 0.06 | 3333.0 | < .001* | 0.24 | 1700.5 | .004* | 0.24 |
| N complaints 'sometimes' & 'often/always' | 13589.5 | .965 | 0.02 | 5332.5 | .044* | 0.13 | 1798.5 | < .001* | 0.32 | 9443.0 | .040* | 0.11 | 3107.5 | < .001* | 0.28 | 1758.0 | .004* | 0.23 |
| N complaints 'often/always' | 12283.0 | .363 | 0.00 | 5523.5 | .087 | 0.11 | 1754.5 | < .001* | 0.35 | 9586.0 | .047* | 0.11 | 2930.0 | < .001* | 0.32 | 1569.5 | < .001* | 0.30 |

*Note*: H&Y = Hoehn and Yahr staging [15]; M = mean; n = number; PD = Parkinson's disease; SD = standard deviation.

* = significant p-value (α < .05).

**Table 7. SVCq scores of male and female individuals with PD, with Mann-Whitney U test results.**

|  | Males with PD (n = 354) | | Females with PD (n = 227) | | | | |
|---|---|---|---|---|---|---|---|
|  | M ± SD | Median | M ± SD | Median | U | p | r |
| Total SVCq score | 8.09 ± 7.30 | 6.00 | 8.95 ± 7.21 | 7.00 | 36612.5 | 0.070 | 0.08 |
| Impact on daily life score | 3.14 ± 2.64 | 3.00 | 3.20 ± 2.75 | 2.00 | 38417.5 | 0.886 | 0.01 |
| N complaints 'sometimes' & 'often/always' | 6.14 ± 4.69 | 5.00 | 6.73 ± 4.62 | 6.00 | 36957.5 | 0.102 | 0.07 |
| N complaints 'often/always' | 1.93 ± 3.06 | 0.00 | 2.17 ± 3.01 | 1.00 | 36345.5 | 0.039* | 0.09 |

*Note*: M = mean; n = number; PD = Parkinson's disease; SD = standard deviation.

* = significant p-value (α < .05).

underdiagnosed in people with PD [4]. In controls, we only had access to self-reported information. Therefore, there is a risk of underestimation in both groups. Conclusions regarding the analyses with OCs should therefore be drawn with caution.

Besides OCs, there might be other factors relating to the presence of visual complaints, which might be subject of future studies. There may for example be detectable visual function deficits associated with complaints. Common complaints in our study were unclear vision, trouble focusing, and difficulty reading. These may be associated with oculomotor deficits. Previous research shows that oculomotor deficits occur regularly in people with PD and influence clarity of vision, the ability to focus, and reading [28]. In addition, people with PD with oculomotor deficits are reported to experience double vision and difficulty driving [28], complaints that were, relative to controls, also very common in PD in our study. Moreover, complaints that may be less dependent on precise eye movements, such as visual field loss and altered color vision, we found to be least common.

However, as was previously shown by others in visual and cognitive domains (Van der Feen et al., submitted) [29], functional deficits might also not strongly relate to the presence of complaints. There may not be a clear one-to-one relationship between complaints and objective functional deficits. In addition, the pathophysiology of PD, including disturbances in retinal, cortical or thalamic functioning, may relate to visual complaints as well [1]. This is supported by our finding that longer disease duration and increasing severity of PD resulted in more visual complaints. Furthermore, as was also shown by previous studies, we found that the amount of antiparkinsonian medication related to the presence of visual complaints as well [30–32].

## Clinical implications

It is highly important to screen for a large range of visual complaints, since each complaint might have its own negative impact on the daily lives of people with PD. Furthermore, each

**Table 8. Spearman's correlations between SVCq scores and age, disease duration, and LEDD of people with PD.**

|  | Age | Disease duration | LEDD |
|---|---|---|---|
| Total SVCq score | r = .118, p = .004* | r = .213, p = < .001* | r = .242, p = < .001* |
| Impact on daily life score | r = .103, p = .014* | r = .236, p = < .001* | r = .235, p = < .001* |
| N complaints 'sometimes' & 'often/always' | r = .102, p = .014* | r = .195, p = < .001* | r = .233, p = < .001* |
| N complaints 'often/always' | r = .141, p = .001* | r = .220, p = < .001* | r = .224, p = < .001* |

*Note*: LEDD = Levodopa equivalent daily dose; n = number; PD = Parkinson's disease.

* = significant p-value (α < .05).

complaint might require another type of treatment. By addressing complaints that are most bothersome for that particular person, the negative impact on the daily lives of people with PD can be reduced to a minimum.

When a person indicates having visual complaints, possible underlying OCs and visual functions may be assessed to identify and, if possible, treat them. Some underlying disorders might be well treated (e.g., dry eyes or cataract) and relieve complaints. However, not all OCs can be treated (e.g., in the presence of oculomotor deficits). Moreover, it may be that complaints not always directly relate to an underlying OC or functional disorder, and relate to the presence of PD itself, or factors related to it (e.g., medication). When this is the case, advice, aids or rehabilitation is required to reduce complaints (e.g., use of a fixed reading distance in case of oculomotor deficits).

Regardless of the cause, timely recognition of visual complaints is important. As PD is a progressive and fluctuating disease, frequent mapping of visual complaints is advised. This is supported by our finding that age, disease duration and severity relate to the occurrence of visual complaints. Since a person might not report visual complaints themselves, standardized questioning is advised. Especially considering that a large proportion of people with PD (42.3%) indicated that they would appreciate advice or rehabilitation for their visual complaints.

When using the SVCq in clinical practice, one could use total or subscale scores to get a clear and general impression of the visual complaints a person experiences, a method supported by our previous study [13]. Additionally, this study illustrates that it may be important to look at individual items of the questionnaire as well. Complaints are not equally common (e.g., we see that dry eyes are more common than painful eyes from the same subscale) and may be distributed differently among individuals. By taking individual complaints into account, care can be tailored specifically to each individual.

## Strengths, limitations and recommendations for future research

This is one of few studies aiming at improving the knowledge on visual complaints in people with PD. Most previous studies on visual complaints included relatively small groups or used measures that captured only one or a few complaints [2]. By including a large cohort of people with PD, and an age-matched control group, we were able to draw conclusions more representative for the whole population. In addition, we were able to examine a wide variety of visual complaints by using the SVCq.

Other screening questionnaires, like the 25-list-item version of the National Eye Institute Visual Function Questionnaire (NEI-VFQ-25) and the more recently developed Visual Impairment in Parkinson's Disease Questionnaire (VIPD-Q) primarily aim to detect vision-related quality of life or symptoms of OCs, respectively [23,33]. In contrast, the SVCq is aimed at assessing a broader range of visual complaints at a functional and activity level and can be used as a guide for treatment or rehabilitation, by focusing on a person's priorities, regardless of the presence of OCs. We do recognize the possibility that OCs or visual function deficits may relate to complaints, as was also suggested by results of our study. However, this study also showed the possibility of complaints occurring in the absence of an identified OC. This underlines the added value of screening for self-reported complaints as well.

Another difference with previous questionnaires is that the SVCq starts with an open-ended question. This allows people to report complaints that are most important to them, without being influenced by the content or direction of the question itself. This seems a valuable addition for rehabilitation purposes. Firstly, it gives an indication of which complaints may be most prominent or relevant to a person, which guides intervention goals. Secondly, it

may provide additional information on complaints that are not (fully) covered by structured items of the questionnaire (e.g., a person may have a complaint especially when they are tired, or during a specific time of the day). Although the open-ended question is a valuable addition, it should not be seen as a substitute for the structured items, as our study showed that people with PD may not report complaints spontaneously, whereas they do when specifically asked. Moreover, based on complaints reported on the open-ended question, it appears that the most prevalent spontaneously reported complaints of people with PD are covered by the structured SVCq items, providing additional supporting evidence for the validity of the questionnaire.

A possible limitation of this study is that data of people with PD were collected in an outpatient clinic, which is not visited by people that are bedridden. Therefore, people with PD in the highest disease stages were underrepresented in this study. Because the prevalence of visual complaints is related to disease stage, our prevalence of complaints may have been underestimated. Nonetheless, this study provides clear insight into the complaints of an outpatient population.

The method of data collection differed for the two groups. Whereas people with PD had the choice of how to complete the questionnaire (online or on paper), controls completed the questionnaire online. In the online setting, people could not continue with the questionnaire until they had given an answer. We were unable to control for this when people completed the questionnaire on paper, which resulted in some missing or unclear data in the group of people with PD. However, missing data was few and randomly distributed. We do not expect that different collection methods led to different response patterns [34].

It cannot be ruled out that comorbidities (e.g., neurological or psychiatric conditions) could have explained part of the complaints experienced by people with PD. For example, stroke is likely to influence vision [35]. Comorbidities were allowed in the PD group, while severe disorders (like a stroke) were excluded in the control group. By allowing comorbidities, however, we did create a representative group of people with PD. Furthermore, we included a large sample, which leads to an accurate estimation of visual complaints in the population. Future studies could consider investigating the influence of other comorbidities on visual complaints in PD.

We had no access to information such as ethnicity and socioeconomic status, or symptoms of anxiety and depression, or cognitive impairment in either group. Since these variables might influence the presence and the extent to which people report visual complaints, this could be subject to future research [36–38].

## Conclusion

People with PD experience a wide range of visual complaints, which occur more frequently in people with PD compared to controls. These complaints seem to evolve with progressing disease characteristics and medication use. Since visual complaints can have a vast impact on the daily lives of people with PD, standardized questioning is advised for timely recognition and treatment.

## Supporting information

**S1 Table. Ophthalmological conditions in people with PD and age-matched control.**
(PDF)

**S2 Table. Screening visual complaints questionnaire–Dutch version.**
(PDF)

**S3 Table. Screening visual complaints questionnaire–English version.**
(PDF)

## Acknowledgments

Research support: E.A. van de Klundert (BSc), Psychology student at the University of Groningen.

## Disclosure

Data from the samples included in this study have been used in a previous study to answer different research questions (I. van der Lijn, G.A. de Haan, F.E. van der Feen, F. Huizinga, A.B. M. Fuermaier, T. van Laar, J. Heutink, The Screening Visual Complaints questionnaire (SVCq) in people with Parkinson's disease—Confirmatory factor analysis and advice for its use in clinical practice, PLoS One. 17 (2022) 1–14. https://doi.org/10.1371/journal.pone. 0272559) [13].

## Author Contributions

**Conceptualization:** Iris van der Lijn, Gera A. de Haan, Fleur E. van der Feen, Famke Huizinga, Catharina Stellingwerf, Teus van Laar, Joost Heutink.

**Data curation:** Iris van der Lijn, Gera A. de Haan, Teus van Laar.

**Formal analysis:** Iris van der Lijn.

**Investigation:** Iris van der Lijn.

**Methodology:** Iris van der Lijn, Gera A. de Haan, Fleur E. van der Feen, Famke Huizinga, Teus van Laar, Joost Heutink.

**Project administration:** Iris van der Lijn.

**Supervision:** Gera A. de Haan, Teus van Laar, Joost Heutink.

**Writing – original draft:** Iris van der Lijn.

**Writing – review & editing:** Iris van der Lijn, Gera A. de Haan, Fleur E. van der Feen, Famke Huizinga, Catharina Stellingwerf, Teus van Laar, Joost Heutink.

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
