## [Decision Letter · Decision Letter 0]

11 Oct 2022

PONE-D-22-24932Prevalence and nature of self-reported visual complaints in people with Parkinson’s disease - Use of the Screening Visual Complaints questionnairePLOS ONE

Dear Dr. van der Lijn,

Thank you for submitting your manuscript to PLOS ONE. After careful consideration, we feel that it has merit but does not fully meet PLOS ONE’s publication criteria as it currently stands. Therefore, we invite you to submit a revised version of the manuscript that addresses the points raised during the review process.

We look forward to receiving your revised manuscript.

Kind regards,

Thiago Fernandes, Sp. Neuro, EbS, PhD

Academic Editor

PLOS ONE

Journal Requirements:

2. We noted in your submission details that a portion of your manuscript may have been presented or published elsewhere. 

" Parts of the same data were used for publication of our research article "The Screening Visual Complaints questionnaire (SVCq) in people with Parkinson’s disease - Confirmatory factor analysis and advice for its use in clinical practice". This does not constitute dual publication, since the earlier publication focused on the validation and use of the questionnaire in a clinical population. In contrast, the current submission uses the questionnaire scores of people with Parkinson's disease with the aim of providing insight into the prevalence of their visual complaints and contributing risk factors."

Additional Editor Comments:

Reviewers have now commented on your study. You will see that we require some amendments to be made before we are able to proceed. 

If you are prepared to undertake the work required, I would be pleased to forward for publication. For your guidance, reviewers' comments are appended below. 

Most of the comments were very quick-to-solve and, overall, we find your study very interesting. You also will notice that some of the comments are really specific, so please be cautious when arguing & making the necessary amendments. 

Please respond to the Reviewer's comments in a separate letter AND highlight all changes in the manuscript. 

Reviewers' comments:

Reviewer's Responses to Questions

**Comments to the Author**

1. Is the manuscript technically sound, and do the data support the conclusions?

Reviewer #1: Yes

Reviewer #2: Yes

Reviewer #3: Yes

Reviewer #4: Partly

Reviewer #5: Partly

2. Has the statistical analysis been performed appropriately and rigorously? 

Reviewer #1: Yes

Reviewer #2: Yes

Reviewer #3: Yes

Reviewer #4: Yes

Reviewer #5: Yes

3. Have the authors made all data underlying the findings in their manuscript fully available?

Reviewer #1: Yes

Reviewer #2: Yes

Reviewer #3: Yes

Reviewer #4: Yes

Reviewer #5: Yes

4. Is the manuscript presented in an intelligible fashion and written in standard English?

Reviewer #1: Yes

Reviewer #2: Yes

Reviewer #3: Yes

Reviewer #4: Yes

Reviewer #5: Yes

5. Review Comments to the Author

Reviewer #1: The fact that visual complaints cannot be fully explained by ophthalmological conditions is not a surprise. Several diseases affecting the central nervous system cause visual disturbances that cannot be detected during the standard ocular and visual examinations. Therefore, research addressing visually guided behaviors and self-reported visual performance is quite relevant in patients with progressive diseases affecting the brain,

It was interesting to learn from the Authors that PD patients are at high risk of visual complaints such as trouble to reading and focusing. These alterations are likely associated, at least partially, with oculomotor impairments since other visual attributes that may be less dependent on precise fixational eye movements, such as visual field and color vision, showed much lower prevalence.

I would like to congratulate the Authors for their great work and well-organized report. I recommend to include in the discussion a brief description concerning the above-mentioned possible associations between visual disturbances and oculomotor alterations in PD patients that may be a subject of future investigations.

Reviewer #2: The study entitled "Prevalence and nature of self-reported visual complaints in people with Parkinson’s

disease - Use of the Screening Visual Complaints questionnaire" is well executed. The authors have made a clear investigation and have prepared the questionnaires to include most common visual defects for early diagnosis among PD patients. However, the authors need to clarify few aspects,

1. In the introduction the authors can write more about the visual defects that are reported among PD patients with relevance to their stage of PD

2. Whether any subsequent ophthalmic examinations were carried out on the self-reported patients to confirm the nature of the visual condition, like assessing their visual acuity to further confirm ? unclear vision appears to be a pre-dominant complaint with 23.2% reporting, so having a simple visual acuity test on these patients would have added more value to the data.

3. Did the authors consider analyzing the data from PD cases based on their stage of the disease apart from classifying based on their age, if so how was the correlation between the visual condition and stages ? This would provide some insight on the ophthalmic deterioration with relevance to the severity of the PD.

4. How the co-morbidity of diabetes among the PD cases were considered ? and does the control group contains diabetes cases ?

5. The authors need to clarify how many of the PD cases reported here were under dopaminergic medication and also of the total PD cases reported how many were freshly identified to have PD and without taking any dopaminergic medication ?

6. Also it would be nice to record the sleep patterns of the PD patients especially those undergoing dopaminergic treatment and correlate with the visual condition ?

Reviewer #3: The manuscript reports an investigation of visual complaints in a population of Parkinson Disease's patients and compared to age-matched controls. I have some minor points to improve the manuscript's quality.

Abstract

Please indicate the numerical prevalence in the results.

Introduction

The Introduction is short, I suggest a paragraph fastly citing the more important or known visual complaints. It would be informative for a general audience.

Methods

Was there some information about the socioeconomic profile from PD patients and controls? If not, include as limitation.

Results

Figure 1 has no legend in the Y- and X- axis, and the numbers have low contrast. I suggest to include the legends and paint in black the numbers and letters. Why is not indicated the statistical difference?

Discussion

-The methods of data collection in PD patients and controls were different and it is an limitation of the study. The authors should discuss about it.

Reviewer #4: 1. The title can be “Prevalence and …….. - Screening Visual Complaints questionnaire outcome” or "outcome of SV...Q"; instead of “Use of the Screening Visual Complaints questionnaire”.

2. Line 69, please define mild, severe, and moderate condition staging.

3. Your control group had mild PD (line 67)? How is that not affecting your comparison with those having PD?

4. “Dutch-speaking people were invited” implies they can be of any ethnicity speaking dutch. Is ethnicity a factor affecting PD? (DOI: 10.3233/JPD-191763)

5. Are your study participants “people” or are they “patients” of the center?

6. Line 154, mentioning Evdk as a research assistant is unnecessary. “Two researchers” is enough.

7. The authors themselves published a systematic review of “Self-reported visual complaints in PD” in 2022 (DOI: 10.3233/JPD-202324). Moreover, the svc questionnaire you have used is an existing one (with a similar sample size and control group DOI: 10.1371/journal.pone.0272559), used previously and the results are in the systematic review itself. Please explain in detail the need for this study and what additional this study is achieving, that others could not before using the same questionnaire. Basically, how did you overcome the limitations of previous studies (if any).

8. Line 84-85, “In addition, a clear and clinically relevant factor structure has been demonstrated in people with PD” is unclear, please restructure.

9. Although PD patients were previously diagnosed and taken from medical files, please explain the diagnostic criteria of PD and staging and also for the control group.

10. Please explain in detail the ophthalmological examination conducted on the PD and control group.

11. Including people both with and without cataract surgery in the PD and control group, are you introducing any kind of bias or not? (Although you have mentioned excluding them did not alter the results). Please discuss this.

12. Conclusion: “Since visual complaints can have a vast impact on the daily lives of people with PD” Have you tested the QOL in people with PD in this study? Either add a reference or remove the line.

13. Tables 7 and 8 look unnecessary. Adding results is enough.

Reviewer #5: Prevalence and nature of self-reported visual complaints in people with Parkinson’s disease- Use of the Screening of Visual Complaints questionnaire

This paper describes the prevalence of visual complaints in people with PD compared to a healthy control group. This subject is relevant and interesting. It deserves more attention in both research and clinical practice, and I agree with standardized screening for ophthalmological problems and disorders However, I do have a few questions and suggestions that may improve the contribution.

- First, I think both the introduction and discussion need some work. It is clear that this is an underrecognized topic and in need of more study. However, I think that these sections lack a good representation of the work that has already been performed on this subject. The introduction should, in addition, also indicate what this study adds to the current literature.

What is truly new (in my opinion tables 5-8) and how could this help to improve health care? Do we need screening by the ophthalmologist based on this study? And on which ophthalmological diseases? It is difficult to treat only complaints without knowing what the underlying disorders might be. It is stated in the discussion that some complaints (most seen in PD) are possible PD pathology related. This is interesting however there is no explanation on this topic.

- In the discussion section, also a more in-depth discussion of the causes and consequences of the findings and a comparison with previous studies should be made. Now only the rationale that screening is important because of a high prevalence and a potential impact on daily life, is repeated several times in both the introduction and discussion.

- The SVCq is introduced as a questionnaire that screens for visual complaints in people with neurodegenerative diseases. However, the questionnaire was validated in people without neurological conditions. Please explain.

- How was the question on dry eyes established? Dry eyes syndrome is a diagnosis and not typically a visual complain. Dry eyes could lead to symptoms of paradoxically to watery/teary eyes, burning/sandy/itchy eyes and/or blurred vision.

- The conclusion drawn in the discussion that the SVCq is very different from other questionnaires such as the VFQ-25 and the VIPD-questionnaire is not clear to me. When compared especially the VIPD-Q and the SVCq many questions are similar, for example distorted images, hallucinations, painful eye, contrast, color vision, double vision. Also, the frequency never, often etc. is very similar. Could the authors explain what the added value of the SVCq is? I agree that structured questioning of visual complaints should become routine care.

- It seems like the severity of complaints score (on a scale of 1-10) was not used in the analyses. The indications ‘never/ hardly’etc. were used to classify severity. Please explain.

- A limitation of the present study is the lack of objective data on ophthalmological disorders. Even though medical files were studied, one of the problems is that ophthalmological disorders are not recognized (and diagnosed) in PD in clinical practice (also see a recent publication of Borm et al. J Neurol 2022). So, the number of OC’s may have been underestimated. In addition, there may still have been some selection bias because only PD patients visiting the outpatient clinic of a university hospital were included. This is a selected population.

- The authors state following: “Therefore, there is a risk of underestimation in both groups. The percentage of people who indicated having visited an ophthalmologist did not differ between the groups (see Table 1). It is therefore unlikely that results were influenced by missing a large number of ophthalmic conditions in one of the groups.”

This statement confused me, since underreporting of complaints is a reason to screen for visual complaints in PD, especially regarding the results of this study with high prevalence of visual complaints in persons with PD. Then it is paradoxal to assume that persons with PD did visit the ophthalmologist with their complaints earlier. Shouldn’t persons with PD show a much higher rate of visiting an ophthalmologist? Could the authors explain this?

- I would have liked to see analyses on the different age categories. A point of discussion is whether the high number of visual complaints and disorders in PD is a consequence of the disease or related to ‘general ageing’. It would be interesting to distinguish the age categories that were matched and perform the prevalence analyses for these different age categories.

- Were co-morbidities in the PD group considered in the analyses in a way?

- In the discussion it is stated that the high prevalence of visual complaints in people with PD may negatively impact daily life. Wasn’t this also studied here? In general, I find the analyses and conclusions on the clinical impact difficult to interpret; this could be described more clearly and explicitly.

6. PLOS authors have the option to publish the peer review history of their article (what does this mean?). If published, this will include your full peer review and any attached files.

Reviewer #1: **Yes: **Mirella Barboni

Reviewer #2: No

Reviewer #3: No

Reviewer #4: No

Reviewer #5: No

---

## [Author Response · Author response to Decision Letter 0]

17 Nov 2022

We would like to thank the reviewers for their kind attention to our paper and their critical eye. We did our utmost to incorporate the feedback and answer the reviewers' questions as well as possible (see below). Lines mentioned refer to the document including track-changes.

Reviewer #1: 

It was interesting to learn from the Authors that PD patients are at high risk of visual complaints such as trouble to reading and focusing. These alterations are likely associated, at least partially, with oculomotor impairments since other visual attributes that may be less dependent on precise fixational eye movements, such as visual field and color vision, showed much lower prevalence. […] I recommend to include in the discussion a brief description concerning the above-mentioned possible associations between visual disturbances and oculomotor alterations in PD patients that may be a subject of future investigations. 

 We added a paragraph on the possible relationship between the most common complaints and oculomotor deficits in the discussion section (lines 315-323). Furthermore, we stated that it might be interesting for future research to look into the possible relationship between visual complaints and functional disorders (lines 371-373).

Reviewer #2: 

In the introduction the authors can write more about the visual defects that are reported among PD patients with relevance to their stage of PD 

 We added a paragraph discussing what was known about visual complaints from the scientific literature. We also briefly discussed the relationship to certain disease-related data, such as disease severity (lines 54-62).

Whether any subsequent ophthalmic examinations were carried out on the self-reported patients to confirm the nature of the visual condition, like assessing their visual acuity to further confirm ? unclear vision appears to be a pre-dominant complaint with 23.2% reporting, so having a simple visual acuity test on these patients would have added more value to the data. 

 We did not include assessments of visual functions. We agree that possible underlying visual function deficits might have been interesting to look into. We have recommended this for future research (lines 371-373). 

The rational for focusing on complaints in the current study was supported by the following: in some cases, the target of care could be an underlying functional impairment, as for example visual acuity (suggested by the reviewer), which might be addressed by the right glasses. In others, however, underlying functional impairments, as for example oculomotor deficits, cannot be treated well, while a complaint might still be addressed in rehabilitation (e.g., by advising a certain reading distance). There may not be a one-to-one association between objectified visual impairments and subjective visual complaints. We added a few sentences on this topic in the discussion section (lines 340-348). Since the reduction of complaints is the goal of rehabilitation, and knowledge on this topic is still scarce, we chose complaints as the subject of this study. 

Did the authors consider analyzing the data from PD cases based on their stage of the disease apart from classifying based on their age, if so how was the correlation between the visual condition and stages ? This would provide some insight on the ophthalmic deterioration with relevance to the severity of the PD. 

 We did take disease severity into account, by calculating differences in the presence of visual complaints between people in several disease stages, according to Hoehn & Yahr (1998; see Table 6a and 6b). These analyses indeed show that visual complaints seem to increase with disease severity. We did not perform correlations.

How the co-morbidity of diabetes among the PD cases were considered ? and does the control group contains diabetes cases ? 

 Supplementary Table 1 presents the comorbid ophthalmological conditions in both groups. We reported diabetes in case there was proven retinopathy (n = 5 in PD, n = 0 in controls). In addition, this study investigated the relationship between the presence of visual complaints and the presence of ophthalmological conditions, including retinopathy. 

The authors need to clarify how many of the PD cases reported here were under dopaminergic medication and also of the total PD cases reported how many were freshly identified to have PD and without taking any dopaminergic medication ? 

 We added four rows in Table 1, describing how many people with PD were taking the most common forms of antiparkinsonian medication and how many were not taking any medication.

Also it would be nice to record the sleep patterns of the PD patients especially those undergoing dopaminergic treatment and correlate with the visual condition ? 

 We agree that this might be interesting for future research. However, we feel this is beyond the subject of the present study.

Reviewer #3: 

Abstract: Please indicate the numerical prevalence in the results. 

 We included the exact prevalence of complaints in the abstract. 

Introduction: The Introduction is short, I suggest a paragraph fastly citing the more important or known visual complaints. It would be informative for a general audience. 

 We added a paragraph discussing what was known about visual complaints from the scientific literature (lines 54-62).

Methods: Was there some information about the socioeconomic profile from PD patients and controls? If not, include as limitation. 

 We did not have access to information regarding the socioeconomic status of the participants. We included a statement on this in the discussion section (lines 413-416).

Results: Figure 1 has no legend in the Y- and X- axis, and the numbers have low contrast. I suggest to include the legends and paint in black the numbers and letters. Why is not indicated the statistical difference? 

 We adjusted the figure based on these comments.

Discussion: The methods of data collection in PD patients and controls were different and it is an limitation of the study. The authors should discuss about it. 

 We added a paragraph on this limitation in the discussion section (lines 392-398). 

Reviewer #4: 

The title can be “Prevalence and …….. - Screening Visual Complaints questionnaire outcome” or "outcome of SV...Q"; instead of “Use of the Screening Visual Complaints questionnaire”. 

 We have changed the title based on the suggestion given.

Line 69, please define mild, severe, and moderate condition staging.

 We referred to notes e-f below Table 1 for a definition (line 88). If the editor wants us to provide a complete list of conditions excluded in the control group, we are able to provide this list.

Your control group had mild PD (line 67)? How is that not affecting your comparison with those having PD?

 The control group had no PD; we made this more specific in the manuscript (lines 85-86). 

“Dutch-speaking people were invited” implies they can be of any ethnicity speaking dutch. Is ethnicity a factor affecting PD? (DOI: 10.3233/JPD-191763) 

 It is true that people in either group could be of any ethnicity, as long as they speak Dutch. We did not collect data on ethnicity, so we cannot draw conclusion regarding its relationship with visual complaints in PD. We included a statement on this in the discussion section (lines 413-416).

Are your study participants “people” or are they “patients” of the center?

 The people with PD were seen for their disease (i.e., as “patients”) at the Parkinson Expertise Center. In accordance with APA 7th (https://apastyle.apa.org/style-grammar-guidelines/bias-free-language/disability), we do not define these people solely by their disease, which is why we use the term “people with PD”. 

Line 154, mentioning Evdk as a research assistant is unnecessary. “Two researchers” is enough. 

 We removed initials from this sentence (lines 176-177). 

The authors themselves published a systematic review of “Self-reported visual complaints in PD” in 2022 (DOI: 10.3233/JPD-202324). Moreover, the svc questionnaire you have used is an existing one (with a similar sample size and control group DOI: 10.1371/journal.pone.0272559), used previously and the results are in the systematic review itself. Please explain in detail the need for this study and what additional this study is achieving, that others could not before using the same questionnaire. Basically, how did you overcome the limitations of previous studies (if any). 

 The results of the study using the SVCq (2022) are not included in the systematic review mentioned, as the review included studies that were published up to February 5, 2021. Furthermore, the previous study using the SVCq had a different purpose, which was to validate the questionnaire and examine the factor structure of the questionnaire in a sample of people with PD, rather than to investigate the prevalence and nature of visual complaints in this sample, as is the case in the present article. The main advantages of the current study is that it investigated a wide variety of function and activity related visual complaints in a large cohort of people with PD, and compared their results to an age-matched control group. As was found by the review mentioned, only few studies have previously aimed for mapping the prevalence of a number of visual complaints in people with PD. Previous studies have mainly used small sample sizes to draw conclusions on only one or a few complaints, sometimes even without comparing it to controls. These are important limitations that are overcome in this study. 

Line 84-85, “In addition, a clear and clinically relevant factor structure has been demonstrated in people with PD” is unclear, please restructure. 

 We have added a sentence to clarify the meaning of this factor structure (lines 103-105).

Although PD patients were previously diagnosed and taken from medical files, please explain the diagnostic criteria of PD and staging and also for the control group. 

 Diagnosis of PD was done based on the UK Parkinson’s Disease Society Brain Bank Diagnostic Criteria. We added this in the manuscript (lines 84-85). Staging of the disease was reported in Table 1, by means of the commonly used staging by Hoehn & Yahr (1998). The control group consisted of a community sample without PD. Therefore, reporting of diagnostic criteria or disease stage is not applicable. 

Please explain in detail the ophthalmological examination conducted on the PD and control group. 

 We did not perform any ophthalmological assessments. Data on any present ophthalmological comorbidities were recorded from medical files in those with PD, and in those without PD (controls) this was based on self-report (see Supplementary Table 1). This procedure was explained in the manuscript (subheading ‘Determining the presence of ophthalmological conditions’, lines 148-155). 

Including people both with and without cataract surgery in the PD and control group, are you introducing any kind of bias or not? (Although you have mentioned excluding them did not alter the results). Please discuss this.

 We explained this more clearly in the manuscript (lines 154-155). We cannot conclude that cataract surgery relieved all complaints, given the possibility of residual long-term effects or secondary cataracts. Therefore, we considered those who underwent surgery to have an OC. Since there was no difference in results after excluding people who had undergone cataract extraction, we do not expect bias of any kind.

Conclusion: “Since visual complaints can have a vast impact on the daily lives of people with PD” Have you tested the QOL in people with PD in this study? Either add a reference or remove the line. 

 This quote refers to the impact of visual complaints on the daily lives of people with PD, which we investigated through the present study (impact on daily life score, 0 - 10 scale). There is no mention of QOL in the cited quote from the conclusion.

Tables 7 and 8 look unnecessary. Adding results is enough.

 Since reviewer #5 argues these tables contain information not previously studied and reported, we left these tables included in the manuscript.

Reviewer #5: 

First, I think both the introduction and discussion need some work. It is clear that this is an underrecognized topic and in need of more study. However, I think that these sections lack a good representation of the work that has already been performed on this subject. The introduction should, in addition, also indicate what this study adds to the current literature. 

 We added a paragraph in the introduction discussing what was known about visual complaints from the scientific literature and what the current study adds to this (lines 54-62 and 70-80). In the discussion section, we also made more links between our results and previous studies (e.g., lines 299-304 and 315-323). 

What is truly new (in my opinion tables 5-8) and how could this help to improve health care? Do we need screening by the ophthalmologist based on this study? And on which ophthalmological diseases? It is difficult to treat only complaints without knowing what the underlying disorders might be. 

 In particular, screening for a wide range of function and activity related visual complaints in a large group of people with PD is something that has not been done. The use of the SVCq allows for screening for isolated complaints focused on both functions and activities, which provides clear goals for care and rehabilitation. We made the added value of this study more clear (e.g., lines 70-80). As this study also shows, visual complaints are not fully explained by ophthalmological conditions. Complaints may also be present in their absence. In case complaints are present, screening for ophthalmological conditions or functional deficits can be done. However, even in the absence of these deficits, visual rehabilitation can be applied to relieve complaints (e.g., by providing aids or advice for activities that a person has difficulty with). We explained this in more detail in the discussion section (lines 340-348). In general, the current study advocates that regular screening for visual complaints is needed so that they are recognized and people can receive the right advice and rehabilitation, regardless of the cause. 

It is stated in the discussion that some complaints (most seen in PD) are possible PD pathology related. This is interesting however there is no explanation on this topic.

 There is a short explanation on this in lines 327-330. We state that for example PD related disturbances in retinal, cortical or thalamic functioning might contribute to the experience of visual complaints (supported by DOI: 10.1093/brain/aww175). In addition, we showed that complaints increase with disease progression.

In the discussion section, also a more in-depth discussion of the causes and consequences of the findings and a comparison with previous studies should be made. Now only the rationale that screening is important because of a high prevalence and a potential impact on daily life, is repeated several times in both the introduction and discussion.

 We complemented the discussion with more links between our results and previous studies. In addition, we focused more on possible causes and consequences of our findings (e.g., lines 299-304, 315-323, and 340-348).

The SVCq is introduced as a questionnaire that screens for visual complaints in people with neurodegenerative diseases. However, the questionnaire was validated in people without neurological conditions. Please explain.

 Even though developed to screen for visual complaints in people with neurodegenerative disorders, the items in the SVCq are general formulations of visual complaints that anyone can experience, not just people with a neurodegenerative disorder. As a first step, the SVCq was examined in a group of people without these disorders. Subsequently, we tested the found structure of the questionnaire in people with PD. Here we found similar results and clinical merit for use of the SVCq in people with PD (DOI: 10.1371/journal.pone.0272559). This is also explained in lines 102-105.

How was the question on dry eyes established? Dry eyes syndrome is a diagnosis and not typically a visual complain. Dry eyes could lead to symptoms of paradoxically to watery/teary eyes, burning/sandy/itchy eyes and/or blurred vision. 

 All items in the questionnaire were established based on literature (in which “dry eyes” were also regularly questioned; DOI: 10.3233/JPD-202324), expert opinions (e.g., neurologists and ophthalmologists working with neurodegenerative disorders, including PD), and experiences of people with a neurodegenerative disorder, including PD. A detailed description of the questionnaire development can be viewed at DOI: 10.1371/journal.pone.0232232. 

The conclusion drawn in the discussion that the SVCq is very different from other questionnaires such as the VFQ-25 and the VIPD-questionnaire is not clear to me. When compared especially the VIPD-Q and the SVCq many questions are similar, for example distorted images, hallucinations, painful eye, contrast, color vision, double vision. Also, the frequency never, often etc. is very similar. Could the authors explain what the added value of the SVCq is? I agree that structured questioning of visual complaints should become routine care. 

 We tried to clarify this more in the discussion (lines 362-373). Despite several complaints appearing in both questionnaires, the VIPD-Q aims to detect underlying ophthalmological conditions. Questions are therefore for example phrased more specifically (e.g., does a complaint occur in a specific situation?). The SVCq asks about isolated complaints, at both a functional and activity level. As we found that complaints do not always directly relate to ophthalmological conditions, the aim of the SVCq is not to detect these conditions, but to guide care and rehabilitation. 

It seems like the severity of complaints score (on a scale of 1-10) was not used in the analyses. The indications ‘never/ hardly’etc. were used to classify severity. Please explain. 

 We used this score as the ‘impact on daily life score’. Throughout the manuscript, we indicated more clearly where this score was used, by mentioning the 0 - 10 scale.

A limitation of the present study is the lack of objective data on ophthalmological disorders. Even though medical files were studied, one of the problems is that ophthalmological disorders are not recognized (and diagnosed) in PD in clinical practice (also see a recent publication of Borm et al. J Neurol 2022). So, the number of OC’s may have been underestimated. In addition, there may still have been some selection bias because only PD patients visiting the outpatient clinic of a university hospital were included. This is a selected population. 

 We agree that these are possible limitations of the study, which we addressed in the discussion section (lines 387-391 and 399-403). 

The authors state following: “Therefore, there is a risk of underestimation in both groups. The percentage of people who indicated having visited an ophthalmologist did not differ between the groups (see Table 1). It is therefore unlikely that results were influenced by missing a large number of ophthalmic conditions in one of the groups.” This statement confused me, since underreporting of complaints is a reason to screen for visual complaints in PD, especially regarding the results of this study with high prevalence of visual complaints in persons with PD. Then it is paradoxal to assume that persons with PD did visit the ophthalmologist with their complaints earlier. Shouldn’t persons with PD show a much higher rate of visiting an ophthalmologist? Could the authors explain this? 

 Thank you for pointing this out. It is indeed incorrect what was stated here. We chose to omit the last two sentences, but to leave the limitation about the possible underestimation of the number of ophthalmic conditions (lines 399-405).

I would have liked to see analyses on the different age categories. A point of discussion is whether the high number of visual complaints and disorders in PD is a consequence of the disease or related to ‘general ageing’. It would be interesting to distinguish the age categories that were matched and perform the prevalence analyses for these different age categories. 

 In this study, we determined a positive correlation between the presence of visual complaints and age. We have also shown that people with PD experienced more visual complaints than controls. Given that these controls were age-matched to the group of people with PD, it can be concluded that the high prevalence of visual complaints in people with PD cannot be fully explained by ageing. We do agree that analyzing each age category may provide even more insight into possible interaction effects. As we already performed many analyses in this study, all of which include important variables, we do not feel that adding another set of analyses on just one of those variables is justified. Nonetheless, if the editor wants us to add these analyses, we can do that.

Were co-morbidities in the PD group considered in the analyses in a way? 

 We did not correct for comorbidities other than ophthalmological conditions. For this reason, there is a paragraph on the possible effect of this decision in the discussion section (lines 406-412). 

In the discussion it is stated that the high prevalence of visual complaints in people with PD may negatively impact daily life. Wasn’t this also studied here? In general, I find the analyses and conclusions on the clinical impact difficult to interpret; this could be described more clearly and explicitly. 

 This was studied indeed, by the ‘impact on daily life score’ (0-10). Conclusions regarding impact on daily life were based on this score. We indicated more clearly where this score was used throughout the manuscript, facilitating interpretation.

---

## [Decision Letter · Decision Letter 1]

5 Dec 2022

PONE-D-22-24932R1Prevalence and nature of self-reported visual complaints in people with Parkinson’s disease - Outcome of the Screening Visual Complaints questionnairePLOS ONE

Dear Dr. van der Lijn,

Thank you for submitting your manuscript to PLOS ONE. After careful consideration, we feel that it has merit but does not fully meet PLOS ONE’s publication criteria as it currently stands. Therefore, we invite you to submit a revised version of the manuscript that addresses the points raised during the review process.

Thank you for submitting your valuable work.

The reviews, which are insightful and interesting, pointed to some unexplained aspects and the authors were able to clarify them.

By my own reading, the manuscript still needs a bit of refinement, mostly related to generalisation, conciseness and the control of confounding factors. Although this may sound counterintuitive, I am keen to understand authors' claims and keen on reading a refined manuscript.

Briefly, as raised by one of the reviewers, you cannot just make too much speculation of your findings.

As we have observed through this time, PD's ocular manifestations are measured by the use of behavioural and physiological techniques. For example, we have noted that the dopamine depletion can affect eye's structure but also functional aspects from the retina to brain (based on ERPs, colour or chromatic discrimination etc.) This is not the focus of the manuscript, indeed, but those findings showed that other measures are reliable enough to detect ocular and visual manifestations, but also help, somehow, during the course of the disease. My concern here is: you are using a self-reported measure to characterise visual complains and, hence, because of that, you cannot just speculate that much with strong statements or even with some assertions. I think the authors can work on a little bit, refining the manuscript to ensure that readers and researchers can easily understand that your claims and assertions are tightened up to your results - without too much "physiology" or "all or nothing" claims.

You will notice that reviewers provided frank and clear comments and, in this round, they almost had no concerns. This is a good point, actually, and the authors can rest assure that this will be considered. However, more important is to provide a frank, clear and transparent perspective of your findings. I'd highly recommend the authors to rethink some statements where sound like "causal" relation of the self-related and visual complains - that should be interpreted as self-reported.

For example, one could say that he/she has no visual complain. But when comes to the physiological aspect, them could have cortical or subcortical damages. Well, by self-report them reported nothing, but physiologically, them have.

So basically I am gentle asking you to be cautious in this aspect and also refine your limitations pointing this difference of self-reported and other measures that really can show causal relation of the complains. Your study has merits and should be considered, but, considering these aspects, please opt to use a more conservative approach for debating your data and for your conclusions.

Overall, the manuscript reads better and has been really refined considering all of the reviewers’ comment. Also I commend the authors for their politeness and efforts. Wishing you success with the study and a very good refined version, so then we can proceed.

We look forward to receiving your revised manuscript.

Kind regards,

Thiago Fernandes, MA, EbS, Sp. Neuro, PhD

Academic Editor

PLOS ONE

Journal Requirements:

Additional Editor Comments:

Please read my comments.

Reviewers' comments:

Reviewer's Responses to Questions

**Comments to the Author**

1. If the authors have adequately addressed your comments raised in a previous round of review and you feel that this manuscript is now acceptable for publication, you may indicate that here to bypass the “Comments to the Author” section, enter your conflict of interest statement in the “Confidential to Editor” section, and submit your "Accept" recommendation.

Reviewer #1: All comments have been addressed

Reviewer #3: All comments have been addressed

Reviewer #4: All comments have been addressed

Reviewer #5: (No Response)

2. Is the manuscript technically sound, and do the data support the conclusions?

Reviewer #1: Yes

Reviewer #3: Yes

Reviewer #4: Yes

Reviewer #5: Partly

3. Has the statistical analysis been performed appropriately and rigorously? 

Reviewer #1: I Don't Know

Reviewer #3: Yes

Reviewer #4: Yes

Reviewer #5: Yes

4. Have the authors made all data underlying the findings in their manuscript fully available?

Reviewer #1: Yes

Reviewer #3: Yes

Reviewer #4: Yes

Reviewer #5: Yes

5. Is the manuscript presented in an intelligible fashion and written in standard English?

Reviewer #1: Yes

Reviewer #3: Yes

Reviewer #4: Yes

Reviewer #5: Yes

6. Review Comments to the Author

Reviewer #1: The discussion regarding the effects of abnormal eye movements on visual functions in PD has been added by the Authors. Thank you!

Reviewer #3: Dear Editor, all my issues were addressed and I considered that the manuscript is suitable to be published

Reviewer #4: (No Response)

Reviewer #5: I would like to thank the authors for addressing all comments of 5 reviewers. Most of my addressed points have been clarified.

Although, I still have concerns about the final conclusion that there are complaints without underlying OC.

In my opinion it is not possible to state this as part of the end conclusion without thorough ophthalmological assessment.

for example impaired color vison could be related to retinal dysfunction this is an OC (due to parkinson's or not) the same for dry eyes, blurred vision due to refractive errors and reading problems due to oculomotor problems or visual field deficitis.

Of course sometimes you don't know what the underlying cause to symptoms is and one can speculate it is parkinson's, but to state this with only self-reported OCs is difficult to accept for me. Especially noticing that PD patients underreport symptoms and therefore not go to eye care as often as they should.

7. PLOS authors have the option to publish the peer review history of their article (what does this mean?). If published, this will include your full peer review and any attached files.

Reviewer #1: **Yes: **Mirella Barboni

Reviewer #3: **Yes: **Givago da Silva Souza

Reviewer #4: **Yes: **Sayantan Biswas

Reviewer #5: No

---

## [Author Response · Author response to Decision Letter 1]

10 Jan 2023

(lines mentioned refer to the document including track-changes)

Reviewer #1: 

The discussion regarding the effects of abnormal eye movements on visual functions in PD has been added by the Authors. Thank you!

 We are pleased that our addition is sufficient. Thank you for the feedback.

Reviewer #3: 

Dear Editor, all my issues were addressed and I considered that the manuscript is suitable to be published

 Thank you for the compliment.

Reviewer #5: 

I would like to thank the authors for addressing all comments of 5 reviewers. Most of my addressed points have been clarified. Although, I still have concerns about the final conclusion that there are complaints without underlying OC. In my opinion it is not possible to state this as part of the end conclusion without thorough ophthalmological assessment. for example impaired color vison could be related to retinal dysfunction this is an OC (due to parkinson's or not) the same for dry eyes, blurred vision due to refractive errors and reading problems due to oculomotor problems or visual field deficitis. Of course sometimes you don't know what the underlying cause to symptoms is and one can speculate it is parkinson's, but to state this with only self-reported OCs is difficult to accept for me. Especially noticing that PD patients underreport symptoms and therefore not go to eye care as often as they should.

 We removed the conclusion on ophthalmological conditions from the abstract and conclusion, as this was not the main aim of our study, and we agree that these conditions were not extensively examined in our study as we used retrospective medical file investigation and self-report (lines 23-39, 425). Since we removed these results from the abstract, we elaborated a bit more on results related to our research questions on the prevalence and nature of visual complaints in people with Parkinson’s disease.

Furthermore, we chose to discuss the possible underestimation of ophthalmological conditions earlier in the discussion section, to enable readers to interpret the results in view of this information. We also emphasized that conclusions regarding analyses with ophthalmological conditions should be made with caution (lines 313-318).

In addition, we replaced ‘ophthalmological conditions’ with ‘identified ophthalmological conditions’ in the discussion section to indicate that we are referring only to what is known to us from medical files and self-reported information. With this, we make it even more clear that we cannot state with certainty that no other conditions were present (lines 310-312, 378).

Editor: 

Thank you for submitting your valuable work.

The reviews, which are insightful and interesting, pointed to some unexplained aspects and the authors were able to clarify them.

By my own reading, the manuscript still needs a bit of refinement, mostly related to generalisation, conciseness and the control of confounding factors. Although this may sound counterintuitive, I am keen to understand authors' claims and keen on reading a refined manuscript.

Briefly, as raised by one of the reviewers, you cannot just make too much speculation of your findings.

As we have observed through this time, PD's ocular manifestations are measured by the use of behavioural and physiological techniques. For example, we have noted that the dopamine depletion can affect eye's structure but also functional aspects from the retina to brain (based on ERPs, colour or chromatic discrimination etc.) This is not the focus of the manuscript, indeed, but those findings showed that other measures are reliable enough to detect ocular and visual manifestations, but also help, somehow, during the course of the disease. My concern here is: you are using a self-reported measure to characterise visual complains and, hence, because of that, you cannot just speculate that much with strong statements or even with some assertions. I think the authors can work on a little bit, refining the manuscript to ensure that readers and researchers can easily understand that your claims and assertions are tightened up to your results - without too much "physiology" or "all or nothing" claims.

You will notice that reviewers provided frank and clear comments and, in this round, they almost had no concerns. This is a good point, actually, and the authors can rest assure that this will be considered. However, more important is to provide a frank, clear and transparent perspective of your findings. I'd highly recommend the authors to rethink some statements where sound like "causal" relation of the self-related and visual complains - that should be interpreted as self-reported.

For example, one could say that he/she has no visual complain. But when comes to the physiological aspect, them could have cortical or subcortical damages. Well, by self-report them reported nothing, but physiologically, them have.

So basically I am gentle asking you to be cautious in this aspect and also refine your limitations pointing this difference of self-reported and other measures that really can show causal relation of the complains. Your study has merits and should be considered, but, considering these aspects, please opt to use a more conservative approach for debating your data and for your conclusions.

Overall, the manuscript reads better and has been really refined considering all of the reviewers’ comment. Also I commend the authors for their politeness and efforts. Wishing you success with the study and a very good refined version, so then we can proceed.

 Thank you for your compliments and feedback. We agree that in a previous version we drew some conclusions that were too strongly formulated. We may have speculated too much, and it was not always clear which results were found in our study and which in previous studies. Also, we may have unintentionally suggested some causal relationships. To correct this, we changed the following (in addition to what was mentioned above in the response to reviewer #5):

- We nuanced statements on complaints being present without ophthalmological conditions, by emphasizing that it is possible that complaints are related to ophthalmological conditions, as this is also suggested by results of our study. As our results raise the possibility that complaints may be present when there is no identified ophthalmological condition, we emphasize that screening for visual complaints is important in conjunction to screening for ophthalmological conditions or functional disorders (lines 375-379).

- We replaced the words "explained" or “contributed to” with "related to" or “associated with”, to indicate that we did not demonstrate any causal relationships with our study (lines 309-337).

- We made it more clear that we did not investigate any other factors besides ophthalmological conditions that may relate to experiencing visual complaints. We raised this as a possible topic for follow-up studies (lines 310-339). 

- We described more clearly which results emerged from our study and which from previous studies (lines 310-339).

- We nuanced statements and conclusions and very tentatively suggest what our study can contribute to the management of visual problems in clinical practice (lines 346-355).

---

## [Editor Report · Decision Letter 2]

12 Jan 2023

PONE-D-22-24932R2Prevalence and nature of self-reported visual complaints in people with Parkinson’s disease - Outcome of the Screening Visual Complaints questionnairePLOS ONE

Dear Dr. van der Lijn,

Thank you for submitting your manuscript to PLOS ONE. After careful consideration, we feel that it has merit but does not fully meet PLOS ONE’s publication criteria as it currently stands. Therefore, we invite you to submit a revised version of the manuscript that addresses the points raised during the review process. Thank you for addressing all comments. As mentioned before, these (10.1371/journal.pone.0272559 & https://doi.org/10.1371/journal.pone.0232232) are studies that could be considered as potential overlapping. I ask the authors to detail in the main text the main differences - with specific details re. the three studies. Also, I request the authors to disclose in a diffrent section of the file, next to the diclosures and/or conflict. 

We look forward to receiving your revised manuscript.

Kind regards,

Thiago P. Fernandes, PhD

Academic Editor

PLOS ONE
---

## [Author Response · Author response to Decision Letter 2]

13 Jan 2023

(lines mentioned refer to the document including track-changes)

Editor: 

As mentioned before, these (10.1371/journal.pone.0272559 & https://doi.org/10.1371/journal.pone.0232232) are studies that could be considered as potential overlapping. I ask the authors to detail in the main text the main differences - with specific details re. the three studies. Also, I request the authors to disclose in a diffrent section of the file, next to the diclosures and/or conflict.

 We agree that it may not have been clear throughout the manuscript that similar datasets have been used previously to answer different research questions. For transparency reasons, we have revised our manuscript, and we hope the revisions clarify that previous studies focused on validating the questionnaire, whereas the current study aims to examine the prevalence of visual complaints in people with Parkinson's disease versus controls, based on the validated questionnaire, but using data from the same samples.

We adjusted the following:

- We cited both previous studies in the introduction, describing what the goals of those studies were as well as the goals of the current study, which do not overlap (lines 72-79). 

- We are transparent about including the same samples in the current study and added a reference to previous studies on any occasion we discuss the samples. However, if you think we are now overdoing the number of references (self-citation), please let us know.

- We added a disclosure to the manuscript stating that data from the samples was published previously. 

We would also like to mention that results of the previous studies were summarized in the method section (subheading ‘Screening Visual Complaints questionnaire’; lines 103-108) and results presented in this study have not been published elsewhere.

---

## [Editor Report · Decision Letter 3]

16 Jan 2023

PONE-D-22-24932R3Prevalence and nature of self-reported visual complaints in people with Parkinson’s disease - Outcome of the Screening Visual Complaints questionnairePLOS ONE

Dear Dr. van der Lijn,

Thank you for submitting your manuscript to PLOS ONE. After careful consideration, we feel that it has merit but does not fully meet PLOS ONE’s publication criteria as it currently stands. Therefore, we invite you to submit a revised version of the manuscript that addresses the points raised during the review process.

I would like to thank you for the edits. To ensure adherence to PLOS policies concerning related studies / overlapping, I’d like to call out authors’ attention to a few another aspects. As observed in the main guidelines (https://journals.plos.org/plosone/s/ethical-publishing-practice#loc-submission-and-publication-of-related-studies), the authors will find:

“If related content is found to be too similar to the PLOS submission, or if a duplicate submission is discovered, we will reject the manuscript.

Duplicate content discovered after publication will be addressed depending on the degree of overlap. The journal may issue a correction or a retraction as appropriate.”

My concern is because you are pointing out that this study has findings on the same sample (PD’s) and the other was a validation (using the same sample). But you may think that others can say that your validation on the sample could be also interpreted as visual complaints. I understand the reasons underlying this, but also understand why to conduct another study, that’s why I think that a few things can be addressed before I continue with your study. I highly recommend the authors to address them (as also informed by the Staff), and my apologies for another round:

Please extend your sample – this will help bring the novel aspects idea. I assume the authors have a few more subjects on any database?Please work on your Title and Methods to make clear: (i) studies won’t be considered as related or overlapped and (ii) your Methods should not be the same as your previous study, so you can link without repetition of the same words or write anotherPlease submit your revised manuscript by Mar 02 2023 11:59PM. If you will need more time than this to complete your revisions, please reply to this message or contact the journal office at plosone@plos.org. Please include the following items when submitting your revised manuscript:A rebuttal letter that responds to each point raised by the academic editor and reviewer(s). You should upload this letter as a separate file labeled 'Response to Reviewers'.A marked-up copy of your manuscript that highlights changes made to the original version. You should upload this as a separate file labeled 'Revised Manuscript with Track Changes'.An unmarked version of your revised paper without tracked changes. You should upload this as a separate file labeled 'Manuscript'.If applicable, we recommend that you deposit your laboratory protocols in protocols.io to enhance the reproducibility of your results. Protocols.io assigns your protocol its own identifier (DOI) so that it can be cited independently in the future. For instructions see: https://journals.plos.org/plosone/s/submission-guidelines#loc-laboratory-protocols. Additionally, PLOS ONE offers an option for publishing peer-reviewed Lab Protocol articles, which describe protocols hosted on protocols.io. Read more information on sharing protocols at https://plos.org/protocols?utm_medium=editorial-email&utm_source=authorletters&utm_campaign=protocols.

We look forward to receiving your revised manuscript.

Kind regards,

Thiago Fernandes

PLOS ONE
---

## [Author Response · Author response to Decision Letter 3]

1 Mar 2023

Dear Dr. Thiago Fernandes,

Enclosed you will find our response to your comments. The most important point we address here is that this paper and the previously published paper on the factor structure of the Screening Visual Complaints questionnaire (SVCq) in PLOS ONE have completely different and equally relevant aims, and perform different analyses, using the same dataset. 

As indicated at the time of initial submission, data used in the current manuscript were previously used for a publication on the construct validity of the SVCq (see doi.org/10.1371/journal.pone.0272559). The current paper builds on the previous one and has used the SVCq to analyse the prevalence of visual complaints in a well-defined PD cohort. Hence, we consider our previous publication as an essential first step, before we could conduct the current analyses on the prevalence of visual complaints. We followed a similar path with our research in people with multiple sclerosis (MS). The first paper was also on the construct validity of the SVCq in people with MS, whereas the second paper published the actual prevalence of visual complaints in people with MS using this SVCq (see doi.org/10.1186/s41687-022-00443-0 and doi.org/10.1016/j.msard.2021.103429). We would like to stress that that dividing these topics over two publications should not be considered “salami slicing”. The tables in the current publication report prevalence figures, whereas our previous publication is discussing which structure of the SVCq would provide the most differentiated and solid outcomes of this scale. 

Since part of your concerns refer to the re-use of data, we would like to address that according to the international guidelines for scientific journals, editors and researchers, the use of previously published data for a new publication is accepted under the following circumstances:

- it should be clear that the dataset has been previously published and reference is made to that previous publication through citations.

- the same dataset is used to answer different research questions.

- research questions are answered with a different statistical method.

- the same dependent and independent variables are used, both in number and in denomination. 

- the results and conclusions clearly complement the previous publication.

Following the above-mentioned international guidelines, we now have clearly indicated that we used the same dataset as our previous PLOS ONE publication, but different parts (subscores) of it, to answer different and complementing research questions (see our Introduction lines 76-83 and Methods section lines 87-88; changes mentioned in this letter refer to the document including track-changes) and results (see Discussion lines 365-371). The main components of the studies are essentially different: the previous study performed a factor analysis on data from people with Parkinson's disease, whereas the current study calculates frequencies of visual complaints in people with Parkinson's disease, compared to controls. We now added in the Method section that analyses and results of the five subscales can be found in our previous PloS One publication (line 199), clarifying what new information the current paper adds to the previous. More importantly, the prevalence of individual visual complaints covered by the SVCq has not been previously published, thus making all results in the current manuscript new. 

Although we appreciate your suggestion to add new data to the current dataset, we do not think this is a good idea. First, adding new data to an existing dataset that has been used for a previous publication could be regarded as augmenting a dataset to mask overlap. The new (augmented) dataset and the previously used dataset would still have the same amount of overlap, albeit a slightly smaller proportion of overlap. Second, the risk that readers and researchers might consider the two datasets to be different is present, and consequently both datasets might be included in a meta-analysis, which would be wrong. A more practical reason is that obtaining new ethical approval, approaching and selecting new participants with Parkinson’s disease from the same population in the North of the Netherlands would take at least a year. 

We regret that we cannot fulfill all of your requests, but we hope that with the changes we have made clear that this paper answers new relevant research questions and is truly an original analysis of a known dataset.

Sincerely yours,

On behalf of Prof. Teus van Laar and Prof. Joost Heutink, 

Iris van der Lijn, MSc, PhD candidate

Royal Dutch Visio & University of Groningen

Department of Clinical and Developmental Neuropsychology 

Grote Kruisstraat 2/1

9712 TS Groningen

The Netherlands

+316-36319906

i.van.der.lijn@rug.nl

---

## [Editor Report · Decision Letter 4]

3 Mar 2023

Prevalence and nature of self-reported visual complaints in people with Parkinson’s disease - Outcome of the Screening Visual Complaints questionnaire

PONE-D-22-24932R4

Dear Dr. van der Lijn,

Thank you for your prompt replies. First, I completely understand authors’ claim – and am not arguing this since these are good points. I called out authors’ attention due to PLOS policies (which I have been informed twice). While I think this is crystal clear, and am proceeding with your study, I just ask in a more friendly way to check again possibility of overlapping in future works using any kind of tool. My “advice” is just some systems will automatically say “no, this is overlapping, do not proceed” and will not let the authors’ explain. As I am sure you are aware of this, just think that my “advice” is because we all went through this kind of situation.

Apart from it, thank you for your explanations & importantly re. the strategy I thought was interesting (add more data). A few times you can bear with this request by others, but I have not seen any issue & think the reply was interesting.

Overall, if there is still any concern or thing remaining, PLOS will contact the authors directly. I am wishing you success with your application (am seeing you are PhD candidate) but also wishing success with your very good study.

We’re pleased to inform you that your manuscript has been judged scientifically suitable for publication and will be formally accepted for publication once it meets all outstanding technical requirements.

Kind regards,

Thiago Fernandes, PhD

Academic Editor

PLOS ONE

Additional Editor Comments (optional):

Thank you for your prompt replies. First, I completely understand authors’ claim – and am not arguing this since these are good points. I called out authors’ attention due to PLOS policies (which I have been informed twice). While I think this is crystal clear, and am proceeding with your study, I just ask in a more friendly way to check again possibility of overlapping in future works using any kind of tool. My “advice” is just some systems will automatically say “no, this is overlapping, do not proceed” and will not let the authors’ explain. As I am sure you are aware of this, just think that my “advice” is because we all went through this kind of situation.

Apart from it, thank you for your explanations & importantly re. the strategy I thought was interesting (add more data). A few times you can bear with this request by others, but I have not seen any issue & think the reply was interesting.

Overall, if there is still any concern or thing remaining, PLOS will contact the authors directly. I am wishing you success with your application (am seeing you are PhD candidate) but also wishing success with your very good study.
---

## [Editor Report · Acceptance letter]

27 Mar 2023

PONE-D-22-24932R4 

Prevalence and nature of self-reported visual complaints in people with Parkinson’s disease - Outcome of the Screening Visual Complaints questionnaire 

Dear Dr. van der Lijn:

I'm pleased to inform you that your manuscript has been deemed suitable for publication in PLOS ONE. Congratulations! Your manuscript is now with our production department. 

Kind regards, 

on behalf of

Dr. Thiago P. Fernandes 

Academic Editor

PLOS ONE